# Monitoring reproduction in cryptic small mammals; using body temperature to identify parturition in an endangered rodent

Timothée Gérard[1,2,*], Hugo Chignec[3], Chantal Poteaux[4], Emilie Long[4] and Caroline Habold[2,*]

## ABSTRACT

As biodiversity continues to decline, accurate monitoring of endangered species reproduction has become essential for informed conservation strategies. However, tracking reproductive events in cryptic species, such as burrowing mammal, can pose serious technical challenges. Reproduction often affects body temperature (T°b) in mammals. In this study, we propose to exploit T°b monitoring as an alternative method to detect parturition events in a cryptic rodent, the common hamster *Cricetus cricetus*. The T°b of hamsters was monitored throughout the reproductive season in controlled conditions (*n*=19 females) and in a mesocosm (*n*=67 females). In the laboratory, females showed elevated T°b from male arrival to the end of pup rearing. T°b characteristically varied on parturition days, when females showed a rapid rise in mean daily T°b (<0.5°C) to levels exciding 38.5°C. The following T°b trend was impacted by the number of pups reared by the mother. In wild like conditions, this pattern allowed us to estimate a total of 123 parturition events. T°b analyses thus provide a reliable mean to estimate litter numbers and birth dates in the wild. This method represents an efficient complementary tool for monitoring reproduction in the common hamster, that could be easily extended to other cryptic mammals.

KEY WORDS: Reproduction monitoring, Parturition, Body temperature, Reproductive success

## INTRODUCTION

Animal species are declining at an alarming rate, highlighting the urgent need for improved monitoring and conservation efforts (Arora, 2018; Jaureguiberry et al., 2022). A key component of wildlife population dynamics is the phenology of reproduction, which strongly influences both juvenile and parental survival, and therefore critically affects population viability (Forrest and Miller-Rushing, 2010). This relationship becomes particularly evident in the context of climate change, which increasingly generates mismatches between reproductive timing and resource availability (Iler et al., 2021; Visser

[1]School of Ecology, Hainan Province Key Laboratory for One Health, Hainan University, 570228, Haikou, China. [2]Département d'Ecologie, Physiologie Ethologie, Institut Pluridisciplinaire Hubert Curien, UMR 7178 IPHC (University of Strasbourg, CNRS), 67200 Strasbourg, France. [3]Laboratoire de Biométrie et Biologie Évolutive, Unité Mixte de Recherche 5558, Centre National de la Recherche Scientifique, Université Lyon 1, 69622 Villeurbanne, France. [4]Laboratoire d'Ethologie Expérimentale et Comparée, UR 4443, Université Sorbonne Paris Nord, 93430 Villetaneuse, France.

*Authors for correspondence (timothee.gerard@iphc.cnrs.fr; caroline.habold@iphc.cnrs.fr)

T.G., 0009-0004-7757-4544; H.C., 0009-0004-7263-910X; C.P., 0000-0002-6815-4650; E.L., 0000-0003-0922-381X; C.H., 0000-0002-6881-6546

and Gienapp, 2019). As such, accurate reproductive data are essential to properly model population dynamic and support conservation measures (Parlato et al., 2021). Accurately characterizing reproductive phenology in viviparous species, such as mammals, requires the ability to identify parturition events. However, parturition can be difficult to observe directly, as it often occurs in natural shelters, nests or burrows (Gonyou and Stookey, 1987; Wolff, 2007). Several methods have been proposed to detect parturition in such case (e.g. GPS data in foxes *Vulpes vulpes*, Walton and Mattisson, 2021, or accelerometers in ewes *Ovis aries*, Gurule et al., 2021). Yet these approaches may suffer from substantial uncertainty and often rely on biologgers (i.e. monitoring devices) whose size is unsuitable for small mammals (McMahon et al., 2011; Rensel, 2025). Consequently, alternative methods for reliably detecting parturition in cryptic small mammals, that are hard to detect because of their elusive behaviour, are greatly needed.

A promising approach to estimate parturition dates involves identifying parturition-specific patterns in body temperature variation (hereafter referred to as T°b). In mammals, reproduction induces significant disruption in T°b regulation (e.g. Fewell, 1995; Kadzere et al., 2002; Laburn et al., 1992; Trethowan et al., 2017). These disruptions appear to be species-specific and may reflect differences in thermoregulatory strategies and thermal environments. For example, during gestation, lions exhibit hypothermia, while moose show hyperthermia, potentially protecting foetuses from thermal stress in hot and cold environments, respectively (*Alces alces*, Græsli et al., 2022; *Panthera leo*, Trethowan et al., 2017). T°b alterations have been documented in a variety of large mammals, particularly in livestock (Diaz et al., 2025; Græsli et al., 2022; Kadzere et al., 2002; Kim et al., 2021; Laburn et al., 1992; Trethowan et al., 2017; Vieira et al., 2020). Variations in T°b throughout reproduction have also been described in small mammals: in mice (*Mus musculus*, Smarr et al., 2016), rats (*Rattus Norvegicus*, Fewell, 1995), dwarf hamster (*Phodopus sp.*, Scribner and Wynne-Edwards, 1994), naked mole rats (*Heterocephalus glaber*, Urison and Buffenstein, 1995) and ground squirrels (*Urocitellus parryii*, Williams et al., 2011), where animals show a decrease of T°b in the end of gestation followed by a sudden increase at parturition. Nevertheless, some aspects, such as the relationship between litter size and T°b variation, remain to been investigated. Additionally, the application of these findings for wildlife monitoring and conservation purposes warrants further consideration.

Here, we propose using T°b data to detect parturition events in a cryptic, endangered and protected rodent, the common hamster (*Cricetus cricetus*). This farmland-dwelling species reproduces from March to September, between hibernation periods (Hufnagl et al., 2011). Because reduced reproductive performances is a major driver of its population decline, this species' reproductive biology has been studied for more than a decade, including in natural conditions (Surov et al., 2016). Common hamsters can produce up to three (occasionally four) litters per year, with births occurring in

burrows that reach depths of up to 2 m (Gérard et al., 2025; Nechay et al., 1977). Parturition occurs after a gestation period of 18 to 20 days (Nechay et al., 1977). Females provide parental care for 3-5 weeks before the weaned pups leave the burrow to disperse (Pluch et al., 2013). Because common hamsters exhibit post-partum oestrus (i.e. females can be fertilized right after parturition, Nechay et al., 1977), pup rearing and gestation can overlap, and the interval between two litters can be as short as 18 days (Gérard et al., 2025). Current methods for estimating parturition events in the wild rely on detecting sudden changes in the body mass of reproductive females (e.g. Franceschini-Zink and Millesi, 2008) or analysing pup body mass at emergence from the burrow (e.g. Fleitz et al., 2024). Both indicators are subject to environmental biases such as population density and resource availability (in mice *M. musculus*; Gerber et al., 2021; Macholán et al., 2012), which can make litter detection ambiguous, especially given the short minimal interval between two litters. Additionally, these methods may fail to detect litters when pups die before weaning. Thus, analysing T°b offers a promising method to refine reproduction monitoring of the common hamster.

To test whether the analysis of T°b can serve as a reliable indicator of parturition in the common hamster, we monitored common hamster's reproduction and the associated T°b variations under laboratory and semi-natural conditions (mesocosm). We hypothesize that: (1) T°b would follow a consistent and unambiguous pattern around parturition, allowing reliable detection of birth events and (2) this T°b pattern would be influenced by litter size, and could potentially be used to estimate it.

## MATERIALS AND METHODS
### Global setup and T°b acquisition
This study focused on female common hamsters, during their reproductive season (April to September) under controlled (laboratory) and semi-natural (mesocosm) conditions. All hamsters originated from the laboratory breeding unit, where hamsters have been bred for eight generations. For T°b acquisition, monitored females were equipped with intraperitoneal iButton temperature loggers (ref. DS1922L, Maxim Integrated) coated with biocompatible bee wax. Coated iButtons weighed 4.0 g ($\approx$1.5% of an adult common hamster). Surgical procedures for implanting and removing loggers followed the protocol described by Weitten et al. (2018). Loggers were removed at the end of reproduction for laboratory monitored animals (8 months after implantation), and after recapture for mesocosm monitored animals (maximum 1 year after implantation). T°b was recorded at a resolution of 0.06°C. To accommodate iButtons' 4096-point memory capacity, the sampling interval was set to 45 min in laboratory, and 135 min in mesocosm.

### Reproduction in controlled conditions
In the laboratory, four females were monitored in 2023 as part of a pilot test, and 15 were monitored in 2025. Hamsters were fed a conventional diet (Top rongeur Guyolap pellets, Evialis, Saint-Nolff, France). Housing and reproduction cages were bedded with corn raid and enriched with PVC refuge boxes, cellulose fibber for nesting and wooden gnawing sticks. Cages were cleaned when the bedding showed signs of sporadic water saturation. Housing units were kept at an ambient temperature of 20°C. Artificial light exposure followed the natural variation of photoperiod at the latitude of Strasbourg, France (48.58° N). To assess T°b variation in non-reproductive females, two were kept unmated. The other 17 females underwent reproduction trials, following the reproductive protocol outlined by Gérard et al. (2024), starting on May 5 of the monitoring year. Nine of the females tested were 1 years old and nulliparous,

while the other eight were 2 years old and had given birth the previous year. The mating pairs were formed to avoid inbreeding. During the pilot test (four females), each mating pair was placed in 380×590×257 mm cages for 7 days. During the main test (16 females), pairs were placed in a set of two 265×420×237 mm cages connected by a cylindrical tunnel (250×100×100 mm) for 14 days. The change in setup aimed to better accommodate the solitary behaviour of hamsters, thus reducing aggression and increasing the chances of successful fertilisation. In both cases, hamsters had access to individual PVC refuge boxes as well as multiple food and water sources. After male removal, females' cages were visually checked every day, between 9 and 10 am, and litter presence was recorded. If no parturition occurred within 21 days, the trial was classified as unsuccessful. If parturition occurred, litter size at birth was recorded, and pups were left with the mother for 35 days before being weaned and weighed. During maternal care, females were provided with food enrichment (mix of maize, wheat and sunflower seeds and puppy kibble). Following pups weaning or failed trial, females entered a new reproduction trial with a different male. Each female underwent two or three reproduction trials per season.

### T°b data processing
Female T°b variation were processed into analysable metrics on a daily basis. T°b data were averaged over 24-h periods (from midnight to midnight, as proposed by Fewell, 1995 and Williams et al., 2011) to obtain daily means (hereafter μT°b) and standard deviations (hereafter T°b SD). Daily variation in T°b was defined as the difference between consecutive days ($\Delta$T°b=μT°b$_n$ − μT°b$_{n-1}$). For each day, the trend in T°b across the preceding and following 7 days was computed and referred to as βT°b pre and βT°b post, respectively. Because shallow torpors strongly bias $\Delta$T°b, βT°b pre and βT°b post, days with μT°b lower than 36°C were excluded from the analyses. These computed daily T°b metrics are illustrated for two focal points in green in Fig. 1. They were tested with regards to female's status on each day, that was classified as paired with a male, gestating, giving birth (parturition), rearing pups or none of the above.

After parturition, female's T°b remained elevated before gradually declining back to pre-parturition level. To study this trend, a two-segment linear-regression was fitted to T°b data in the 30 days following parturition. The first segment was constrained to be flat to capture the plateau, and the second was left unconstrained to model the subsequent decline. The breakpoint between the two segments was determined as the one allowing for the best overall fitness of the two-segment linear regression to T°b data (lowest AICc). From this analysis was extracted the mean T°b value during the plateau, the day at which T°b started to decrease (the breakpoint, equivalent to the plateau length), and the rate of decline (expressed in °C per days). Theses parameters are illustrated for two individuals in Fig. 2.

### Reproduction in wild-like conditions
The mesocosm study was conducted in a predator-proof enclosure located near Ittenheim, France. The infrastructure and the monitoring protocol used for mesocosm monitoring are described in Gérard et al. (2025). Studied females were 1 year old and nulliparous at the time of release. They were released on April 19, 2022 (*n*=32), May 22, 2023 (*n*=38), and May 16, 2024 (*n*=32). Males (*n*=12, 15, 12, respectively) were released within 1 week following female release, allowing reproduction to occur freely. Females were recaptured and returned to the laboratory at the end of summer (from September 1 onward) for iButton retrieval. When females could not be recaptured, the temperature loggers and

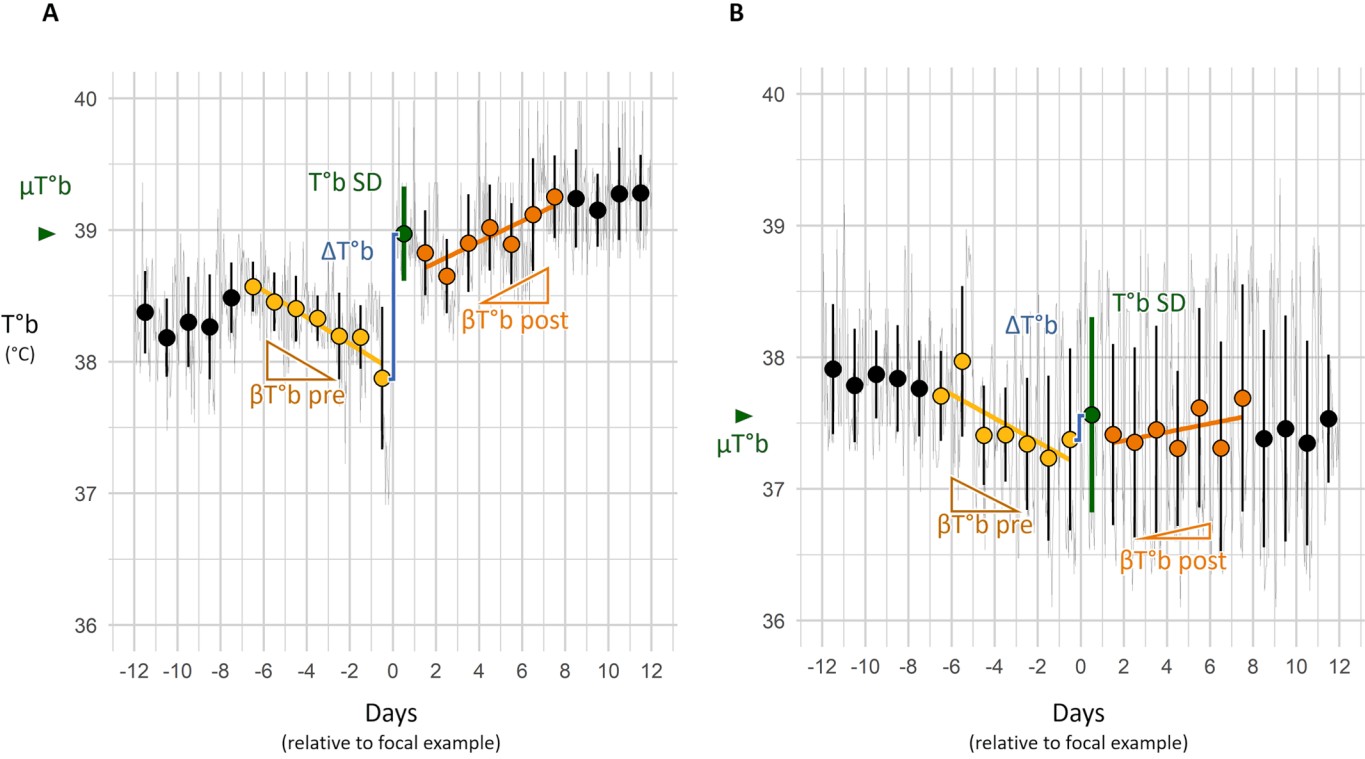

**Fig. 1. Graphs illustrating T°b data processing over two 24-day periods in two females.** The light grey line in the background represents the individual raw T°b data. Points and error bars indicate the daily mean T°b (µT°b) and its standard deviation (T°b SD). As examples, focal points are highlighted in green. Daily variation (ΔT°b) is represented by blue bars. T°b trends over the seven preceding days (βT°b pre) and the seven following days (βT°b post) were calculated from the yellow and orange points, respectively, and illustrated with lines of the corresponding colours. In graph A, the focal point corresponds to a parturition day. In graph B, it corresponds to a randomly chosen day during a period when the female was alone.

associated T°b data were lost. This resulted in a final sample size of 67 females (26 in 2022, corresponding to those presented in Gérard et al., 2025, 23 in 2023 and 18 in 2024). Parturition events were identified from T°b patterns using the criteria selected and presented in the Results section. Because females in the mesocosm could reproduce multiple times within a season, post-parturition T°b analyses were conducted over shortened 20-day periods.

During the reproductive season, weekly trapping sessions provided access to pups once they emerged from their mother burrow. Hair or ear tissue samples were collected from released adults and from trapped pups (under light isoflurane anaesthesia, as detailed in Gérard et al., 2025). These samples were used for DNA extraction and genetic parentage analyses. Parentage assignment was conducted using 15 microsatellite loci, following the protocol of Reiners et al. (2014) and using the Cervus software (v3.0.7, Kalinowski et al., 2007). Details of the analysed microsatellite loci are provided in Table S1. Parentage simulations were based on 10,000 hypothetical offspring, allowing for a 1% genotyping error rate and assuming no unsampled parents. This approach yielded a critical delta in Likelihood-of-Difference (LOD) value of 0.01 for 95% ($n$=589 pups) or 80% ($n$=64 pups) confidence. Pups that could not be assigned with at least 80% confidence ($n$=2) were excluded from the analyses. Females lacking retrievable T°b data were nonetheless included as potential mothers in the parentage analysis. When maternal T°b data were available (496 out of the 655 trapped pups), pups were assigned to a specific litter, and birth date, based on the date and body mass at capture, as illustrated in Fig. 3. Pups captured too early to belong to a second litter were assigned to the first litter (assuming a minimal weaning period of 3 weeks; $n$=108 out of 296 pups in 2022, 93 out of 108 in 2023, 87 out of 91 in 2024). Their

growth rates were then used as a reference to assign the remaining pups to the most probable litter.

## Statistics

Statistical analyses were conducted on *R* (v4.4.3; R Core Team, 2023). Models were computed using the *lme4* package (v1.1.37; Bates et al., 2015), with a significant threshold set at α=0.05. Statistics outputs are detailed in the main text and in the supplementary material. The T°b of laboratory females' according to their reproductive status and age was investigated through linear mixed models (LMM) with the female identity as a random parameter (statistics of these models are detailed in Table S2). Status pairwise comparison was conducted using the Tukey tests from the *multcomp* package (v1.4.28; Hothorn et al., 2008, Table S3). The relationship between laboratory litter size and maternal T°b patterns was studied with generalised linear model (GLM) using Poisson distribution to accommodate the count nature of the response (with no random parameters, as females only had one litter). For the mesocosm data, equivalent analyses were performed with a generalised linear mixed model (GLMM) using Poisson distribution, accounting for mothers' identity and the year of monitoring as random parameters. Graphical representations were designed using the *ggplot2* package (v3.5.1; Wickham, 2016). All data and codes used for this article are available in the online data repository (Gerard et al., 2026).

## Ethics

This study followed the European Directive 2010/63/EU on the protection of animals used for scientific purposes, and were approved by the Ethical Committee (CREMEAS) under APAFIS agreement n°17484-2018103016124862 v3.

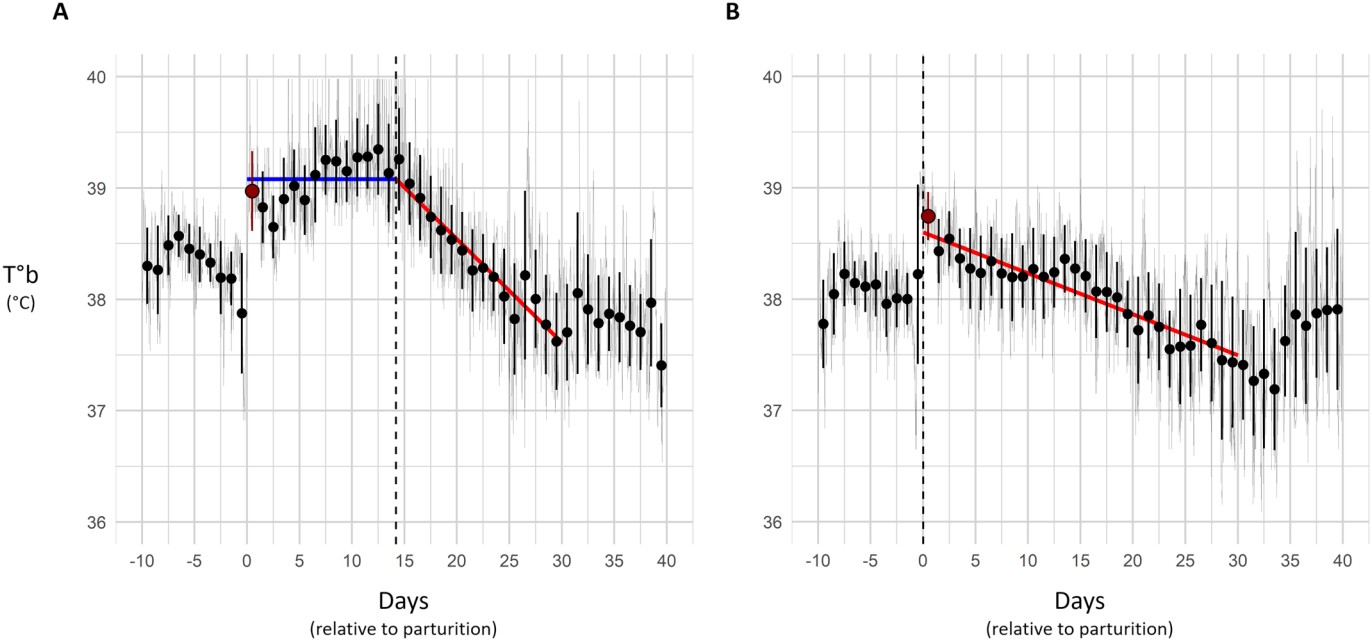

**Fig. 2. Graphs illustrating the T°b trend after parturition of two females.** The light grey line in the background represents the raw individual T°b data. Points and error bars indicate the daily mean T°b (µT°b) and its standard deviation (T°b SD). The days are indexed on parturition (Day 0, in red). The lines represent the two-segment linear regressions during the plateau (blue) and decline (red) of T°b. The dashed line indicates the two-segment linear regression breakpoint (i.e. the end of the T°b plateau). In the case of individual B, no plateau was observed, leading to a plateau duration of 0.

## RESULTS

### Variation of T°b throughout reproduction and parturition

In the laboratory, 15 birth events were observed from nine 1-year-old and six 2-year-old females. Litter size at birth ranged from three to 11 pups (mean: 7.07±2.37 pups per litter), for a total of 106 pups. Seven pups from three different litters died before weaning. All deaths happened in the first 3 days of life. Outside of reproduction, female's mean µT°b were lower in older females (mean 1-year-old=37.438±0.458°C, 2-year-old=37.163±0.523°C, LMM, β=−0.327, d.f.=1, t=−2.609, P=0.009), but their T°b SD were similar (LMM, β=0.004, d.f.=1, t=0.147, P=0.883).

T°b variation in reproducing females throughout reproduction is illustrated in Fig. 4A. For comparison, non-reproducing females are shown in Fig. 4B. T°b metrics according to female status are presented in Table 1. All T°b metrics were significantly impacted by female status (LMM models, d.f.=4, P<0.001 for all T°b metrics, see Table S2 for detailed statistics). Pairwise comparisons among statuses are summarised in Table 1 and detailed in Table S3. Across reproduction, females exhibited an increase in µT°b (+0.609 to 1.51°C depending on status, Tukey, P<0.001 in all cases, see Table 1 and Table S3) and a decrease in T°b SD (reduced by up to 50%, depending on status, Tukey, P<0.001 in all cases,

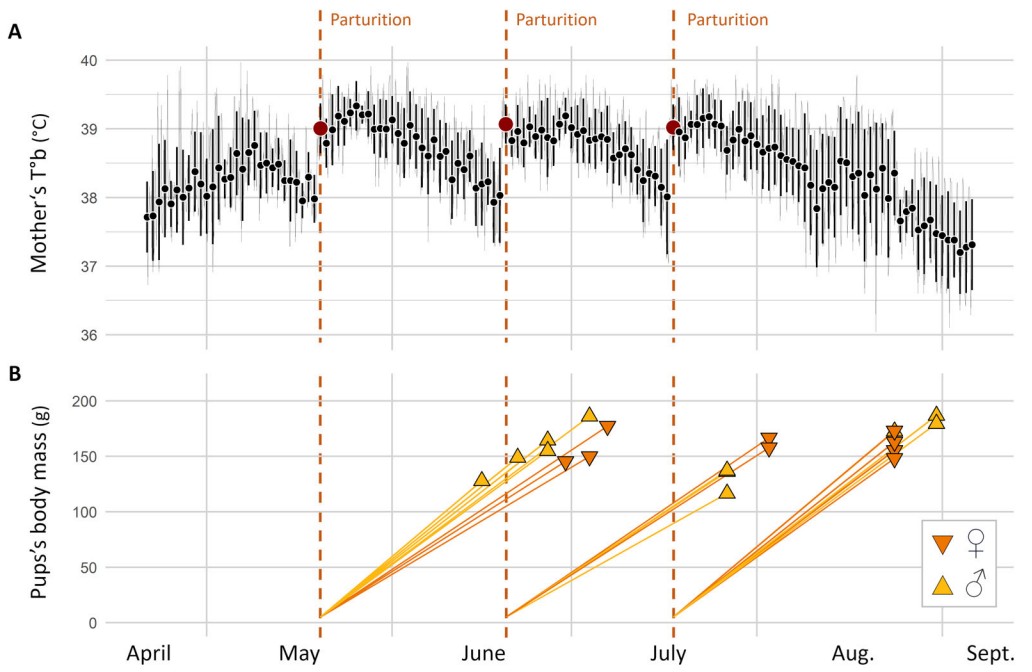

**Fig. 3. Graphs illustrating the use of T°b analyses to determine the birth date of pups born in mesocosm.** (A) T°b variation of a focal mother. The light grey line represents the raw individual T°b data. Points and error bars indicate the daily mean T°b (µT°b) and its standard deviation (T°b SD). Three identified parturition dates are indicated by red points and dashed vertical bars. (B) Mass of the pups of the mother at the date of their capture (n=20). Colours and shapes of the points represent pups' sex, as indicated in the legend. The yellow lines represent each pup's presumed affiliation with a specific date of birth.

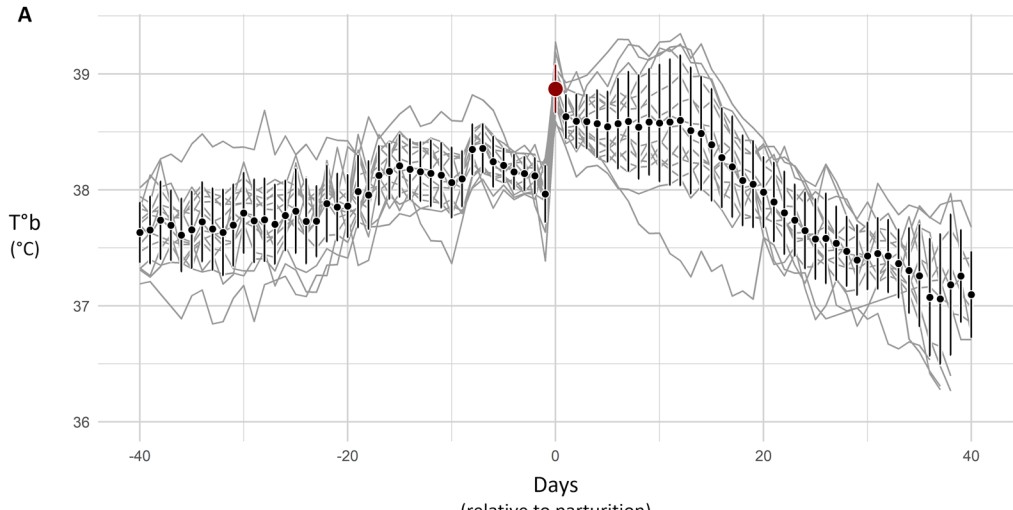

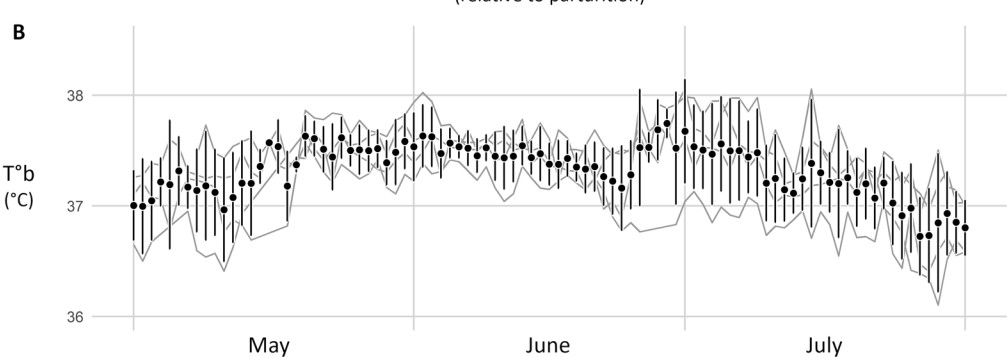

**Fig. 4. Graphs showing the average T°b profile of females along time.** Grey lines represent individual data (μT°b), while points with error bars represent the mean and variance of all females. Graph A shows T°b variation 40 days before and after a parturition. Days were numbered relative to the day of parturition (day 0, in red), $n$=15 females. Graph B shows T°b variation over 3 months for females that had no litter, $n$=4.

see Table 1 and Table S3). The most pronounced changes occurred on the day of parturition, which was characterized by the highest daily amplitude (ΔT°b, +0.906°C, Tukey, $P<0.001$ compared to all other status, see Table 1 and Table S3). In the days preceding and following parturition, T°b showed a decreasing trend (βT°b pre of −0.053°C day$^{-1}$ and βT°b post of −0.008°C day$^{-1}$).

However, the pairwise comparisons showed that only μT°b and ΔT°b changed in a way specific to parturition (LMM-Tukey, $P<0.001$ for all pairs involving parturition, see Table 1 and Table S3 for details). Indeed, T°b SD could be reduced to a similar extent during gestation (Tukey, β=0.067, z=1.931, $P=0.266$), while βT°b pre could be comparably steep during pup rearing (Tukey, β=0.034, z=1.978, $P=0.242$). Thus, μT°b and ΔT°b emerged as the most reliable indicators for detecting parturition. The association between daily μT°b and ΔT°b values and the females' reproductive status is shown in Fig. 5. Under laboratory conditions, days on which μT°b exceeded 38.5°C and ΔT°b was greater than 0.5°C were consistently and exclusively associated with parturition.

### Effect of litter size on T°b variation at parturition and following days

In this study, individual pup's mass at weaning was similar among litters (mean pups' mass; 197.716±13.431 g). Thus, analysis of litter effects was based on litter size, rather than other indicators such as the total mass of pups produced. T°b metrics on the days of parturition were more strongly correlated with litter size at weaning than at birth ($R^2$ birth: 0.353, weaning: 0.597). However, litter size at weaning had no impact on μT°b (GLM, β=0.556, d.f.=1, z=0.869, $P=0.376$), T°b SD (GLM, β=−0.315, d.f.=1, z=−0.184, $P=0.853$), ΔT°b (GLM, β=0.748, d.f.=1, z=1.070, $P=0.288$) or βT°b pre (GLM, β=4.743, d.f.=1, z=1.079, $P=0.266$) on the days of parturition. A higher number of pups at weaning was associated with a higher βT°b post, indicating a prolongation of elevated T°b after parturition (GLM, β=6.746, d.f.=1, z=2.609, $P=0.008$).

This relationship was examined into more details by modelling T°b variation for 30 days following parturition with a two-segment linear model (see Materials and Methods section for details). Here

**Table 1. Empirical values (mean±s.d.) of T°b metrics depending on female condition**

| Female status | Parturition day | None | End of gestation | Rearing pups | Male presence |
|---|---|---|---|---|---|
| **μT°b** (°C) | 38.870±0.203 **a** | 37.326±0.504 d | 38.170±0.233 b | 38.089±0.611 b | 37.935±0.304 c |
| **ΔT°b** (°C) | 0.906±0.283 **a** | −0.016±0.182 b | 0.016±0.162 bc | −0.047±0.142 c | −0.032±0.204 d |
| **T°b SD** (°C) | 0.258±0.074 **d** | 0.515±0.125 a | 0.325±0.131 **d** | 0.386±0.148 c | 0.461±0.149 b |
| **βT°b pre** (°C/day) | −0.053±0.050 **b** | −0.008±0.067 c | 0.003±0.044 c | −0.020±0.070 **b** | 0.036±0.062 a |
| **βT°b post** (°C/day) | −0.008±0.047 **abc** | 0.002±0.104 **a** | 0.033±0.074 **a** | −0.060±0.160 **b** | −0.011±0.051 **c** |

Letters indicate statistically significant T°b metrics differences between categories (Tukey, $P<0.05$, details in Table S3). Letters common with the parturition days (indicating no statistically significant difference) are highlighted in bold.

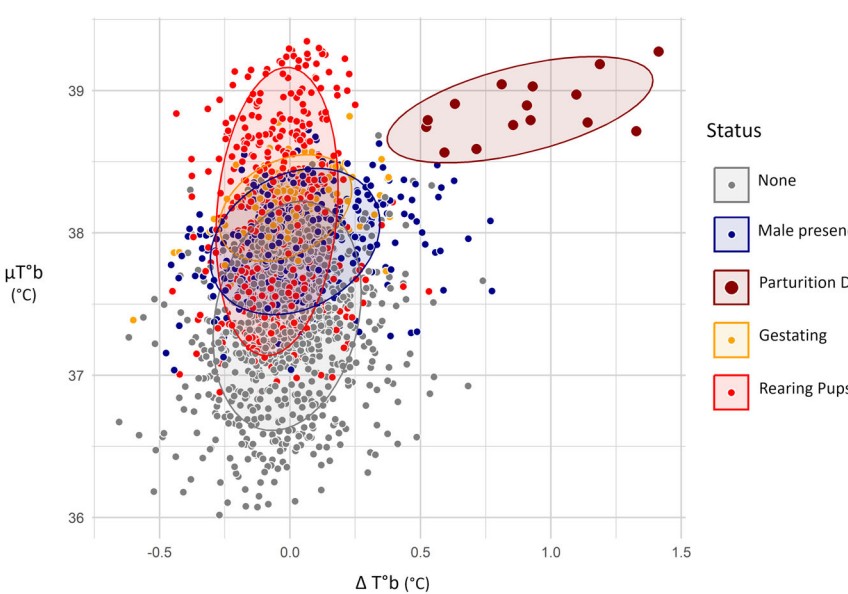

**Fig. 5. Graphs illustrating the link between days µT°b, ΔT°b and females' reproductive status.** Confidence intervals (0.9) are represented as ellipses. Points and ellipses are coloured by status, as indicated by the legend. *n*=2245 days.

also, T°b metrics were more strongly related to the litter size at weaning than at birth (*R²* birth: 0.382, weaning: 0.683). Females that weaned more pups showed a longer T°b plateau (GLM, β=0.103, d.f.=1, z=2.535, *P*=0.004, Fig. 6C), with a higher mean T°b (GLM, β=1.052, d.f.=1, z=2.860, *P*=0.003, Fig. 6B) leading to plateaus of up to 15 days at 39°C. There was no statistically significant link between the number of raised pups and the steepness of T°b

after the plateau (GLM, β=3.323, d.f.=1, z=1.179, *P*=0.254, Fig. 6C).

## Parturition estimation in semi-natural conditions

In the mesocosm, the T°b pattern allowed to identify 123 parturitions over 3 years. Examples of T°b patterns and corresponding litter identifications are presented in Fig. 3 and Fig. 7. In the mesocosm,

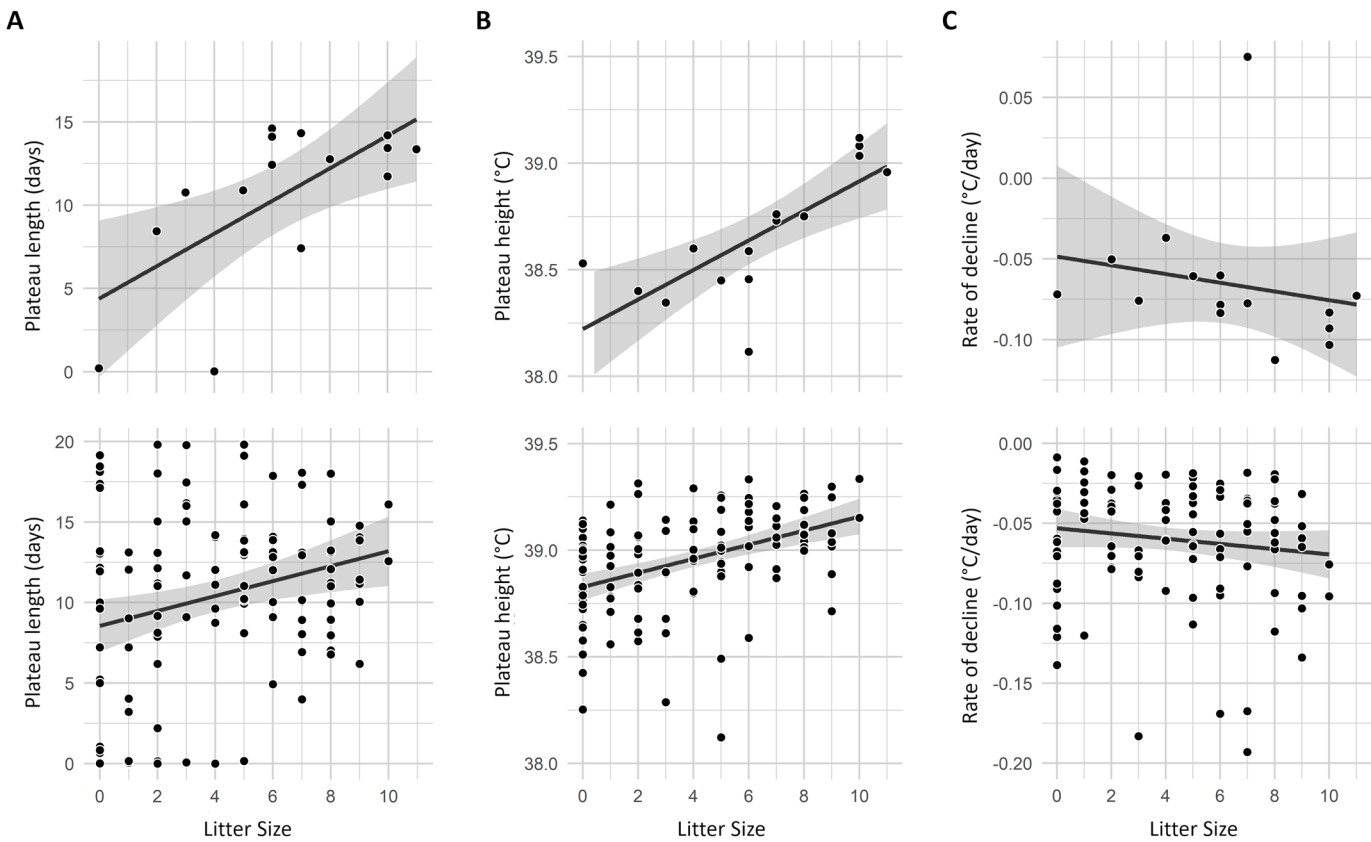

**Fig. 6. Graphs showing the relation between litter size and (A) post-parturition T°b plateau height (°C), (B) post-parturition T°b plateau length (days) and (C) the rate of T°b decline back to baseline levels.** Graphs from the top row depict data from the laboratory monitoring where litter size as weaning was counted directly (*n*=15). Graphs from the bottom row depict data from the mesocosm monitoring where litter size was inferred from genetic and pups mass analyses (*n*=123).

females had up four litters (mean=1.83±0.85 litters). Only one female did not have a litter (panel A in Fig. 7), and no offspring were attributed to her according to the genetic analyses either. The mean litter size at weaning attributed by genetic analysis of the trapped pups was 4.03±3.07 pups. It should be noted that 26 births were observed without any offspring being assigned (litter size=0). Within the enclosure, a higher number of pups in a litter was associated with a longer plateau (GLMM, β=0.046, d.f.=1, z=4.491, P<0.001, Fig. 6A), and a higher T°b (GLMM, β=1.807, d.f.=1, z=6.308, P<0.001, Fig. 6B), but did not impact the return to pre-parturition T°b levels (β=−0.033, d.f.=1, z=−0.815, P=0.415, Fig. 6C).

## DISCUSSION

Common hamsters' parturition induces a specific pattern of T°b, in accordance with our first prediction. At parturition, females showed a sudden increase in T°b (ΔT°b≈+0.9°C, with the lowest observed increases being +0.5°C) reaching a high T°b (μT°b≈38.9°C, with the lowest peak at 38.5°C). This pattern allowed us to identify parturitions under semi-natural conditions. After parturition, T°b remained elevated for a period whose amplitude and duration were proportional to the size of the litter, before gradually declining back to pre-parturition levels. This was coherent with our second prediction. However, the relationship was too weak to predict

litter size based solely on variation in T°b, particularly because this plateau was not systematically observed.

### Reproduction induced T°b variation

Reproduction in female common hamsters strongly affected their T°b. Females had a higher T°b throughout reproduction (in the presence of a male, during gestation and pups rearing). Parturitions were preceded by a decrease of T°b value (βT°b pre≈−0.05°C.day⁻¹) and daily variation (T°b SD, reduced by half compared to normal days, see Table 1 and Fig. 4). Then, parturition was characterised by an important and sudden μT°b increase. This is similar to what was reported in other rodent species, such as *M. musculus* (Smarr et al., 2016), *Rattus norvegicus* (Fewell, 1995), *Phodopus sp.* (Scribner and Wynne-Edwards, 1994) and *Urocitellus parryii* (Williams et al., 2011). This increase could be the result of increased energy expenditure, leading to greater heat production, as suggested by the concomitant variation of T°b and metabolic rate in naked mole rats during reproduction (*Heterocephalus glaber*, Urison and Buffenstein, 1995). Alternatively, the increase of T°b could be linked to an endogenous alteration of T°b to provide heat, while pups are not able to thermoregulate autonomously, as proposed by Scribner and Wynne-Edwards, (1994). In all cases, as small mammals tend to share a common reproductive strategy (large altricial litters that are intensively reared during a few weeks only), it is likely

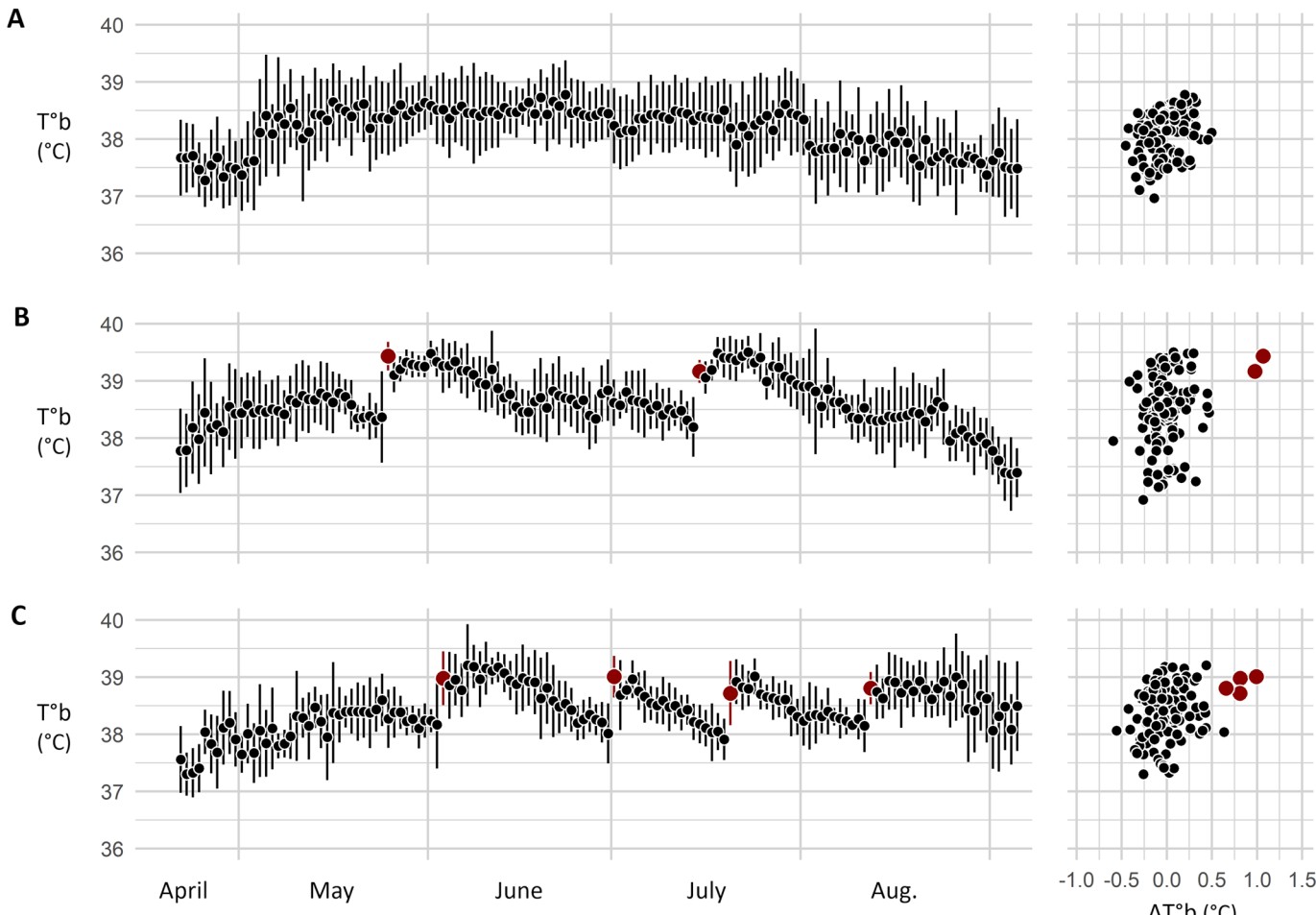

**Fig. 7. Graphs illustrating parturition day identification from the mesocosm monitoring.** On the left, T°b variation throughout the reproductive season of three mothers. Points and error bars indicate the daily mean T°b (μT°b) and its standard deviation (T°b SD). On the left, the μT°b points are plotted against their ΔT°b value, isolating days identified as parturition, depicted in red. In these examples, 0 (A), 2 (B) and 4 (C) parturitions were identified.

that such T°b variation at parturition would be observed in other rodents.

The link between metabolic rate and T°b could explain the impact of litter size on T°b patterns, and the maintenance of high T°b after parturition, during the first weeks of pup rearing. For mammals, lactation is the most energetically demanding part of reproduction (Hayssen and Orr, 2017; Speakman, 2008). This may explain the observed relationship between litter size and T°b plateau height and length (Fig. 6). Furthermore, it is the number of pups at weaning, (i.e. of suckled pups), and not the number of born ones, that most strongly affected these parameters, suggesting a physiological adaptation of females to the energy demand induced by the number of fed pups. This is also supported by the fact that the maximum duration of the plateau observed in females with the largest litters was 15 to 20 days, which corresponds to the age at which common hamster pups start to feed autonomously, thereby reducing the mother's energy requirements (Nechay et al., 1977; Speakman, 2008, personal observation).

### Applicability to the monitoring and conservation of the wild common hamster

In this study, T°b data were obtained using intraperitoneally implanted iButton data loggers. Owing to their internal memory capacity, iButtons eliminate the need for additional equipment. However, their use necessitates two surgical procedures and entails the potential loss of T°b data if animals cannot be recaptured. Alternative loggers may be more suitable depending on the specific monitoring design. Remote-sensing loggers, such as implanted Anipills® (Animals Monitoring, Caen, France), or thermosensitive PIT-tags, can provide higher T°b sampling frequencies, minimise data loss, and reduce animal disturbance by requiring only one or no surgical intervention. Conversely, these loggers necessitate the deployment of antennae for data acquisition, which introduces additional technical constraints. Consequently, studies should optimise their T°b data acquisition protocols by balancing scientific objectives, available technological resources and animal welfare considerations.

Hamsters' T°b varied in a marked and characteristic manner during reproduction. Under laboratory conditions, the concomitant occurrence of a sudden increase in T°b (>0.5°C) and high absolute values (>38.5°C) was never observed outside parturition days (Figs 4 and 5). Consequently, T°b analyses provide a reliable tool for identifying birth events in wild hamsters, as illustrated by our mesocosm monitoring (Figs 3 and 7). However, although T°b analyses provide information on reproduction timing and litter number, they are not sufficient on their own to infer litter characteristics (litter size, sex ratio, pup's growth, etc.), as indicated by the high variability of T°b metrics around the regressions shown in Fig. 6. For a more comprehensive quantification of reproductive success in hamsters, this method can be coupled with complementary approaches, such as pup trapping and genetic analyses. Remarkably, T°b analyses also allowed the identification of failed litters ($n=26$ out of 123), where no pups emerged from the burrow. Litter failure in hamsters is known to occur (Gérard et al., 2024; Surov et al., 2016; Tissier et al., 2017), but it can be difficult or impossible to detect using methods based solely on analyses of mother or pup body mass. In this context, T°b analyses substantially improve the reliability and confidence of parturition detection and reproduction monitoring. Overall, refined reproductive data can be critical in the face of unfavourable wild population dynamics linked to reproduction. T°b analyses can therefore provide key data to support population dynamics modelling, such as the work conducted by La Haye et al. (2014) and Descamps and De Vocht (2025) for the common hamster, or similar approaches applied to other species (Parlato et al., 2021).

### Conclusion

T°b analyses appear to be a useful tool for monitoring reproduction in wild hamsters, particularly when used in combination with other techniques such as pup trapping and genetic analyses. In this and other species, this approach can be especially valuable for precisely quantifying reproductive success and phenology, thereby supporting conservation efforts.

**Acknowledgements**
The authors sincerely thank the DEPE (IPHC-CNRS) animal facility staff – Hélène Gachot, David Bock, Aurélie Hranitzky, and Nicolas Spanier – for their invaluable assistance with animal care and facility management.

**Competing interests**
The authors declare no competing or financial interests.

**Author contributions**
Conceptualization: T.G., C.H.; Data curation: T.G., H.C.; Formal analysis: T.G.; Funding acquisition: C.H.; Investigation: T.G., H.C., E.L.; Methodology: T.G., H.C., C.P., C.H.; Project administration: T.G., C.H.; Validation: T.G., H.C., C.P., C.H.; Visualization: T.G.; Writing – original draft: T.G.; Writing – review & editing: T.G., H.C., C.P., C.H.

**Disclosure on the use of AI**
No artificial intelligence was used to create the original content of the article. ChatGPT (OpenAI, 2025) was used to suggest article phrasing improvements.

**Funding**
This work was funded by the Fonds Vert program of the Direction régionale de l'Environnement, de l'Aménagement et du Logement (DREAL), Grand Est region, France. Open Access funding provided by CNRS - IPHC - DEPE. Deposited in PMC for immediate release.

**Data and resource availability**
All data, code and materials used in this study is available online in the Environment and Society Data Inventory (InDoRES) data repository: https://doi.org/10.48579/PRO/SSSV2T (Gérard et al., 2026). All relevant data and details of resources can be found within the article and its supplementary information.

**First Person**
This article has an associated First Person interview with the first author of the paper.

**Peer review history**
The peer review history is available online at https://journals.biologists.com/bio/lookup/doi/10.1242/bio.062459.reviewer-comments.pdf

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
