## [Peer Review File · Biology Open]

Monitoring reproduction in cryptic small mammals; Using body temperature to identify parturition in an endangered rodent

Hugo Chignec, Chantal Poteau, Emilie Long, Caroline Habold and Timothée Gérard

DOI: 10.1242/bio.062459

Editor: Lewis Halsey

Review timeline

Original submission:	25 December 2025
Editorial decision:	7 January 2026
First revision received:	12 March 2026
Accepted:	19 March 2026

Original submission

First decision letter

MS ID#: bio.062459

MS Title: Monitoring reproduction in cryptic small mammals; Using body temperature to identify parturition in an endangered rodent

Authors: Timothée Gérard, Hugo Chignec, Chantal Poteau, Emilie Long and Caroline Habold

I have now reached a decision on the above manuscript.

The reviewer reports are shown at the bottom of this email.

As you will see, the reviewers raised a number of substantial criticisms that prevent me from accepting the paper at this stage.

They suggest, however, that a revised version might prove acceptable, if you can address their concerns. If you think that you can deal satisfactorily with the criticisms on revision, I would be pleased to see a revised manuscript. We would then return it to the reviewers.

At this stage, we also ask you to ensure your manuscript complies with our formatting guidelines. Provided you are able to fully address the referees' comments, we are positive about publication of your paper (we accept over 95% of revision submissions) and therefore hope you won't mind any extra work involved in reformatting your manuscript at this point.

Please upload both a 'clean' version of your Word file, along with a highlighted version clearly showing where you have made changes in the revised manuscript. Please avoid using 'Track changes' in Word files as these are lost in PDF conversion.

I should be grateful if you would also provide a point-by-point response detailing how you have dealt with the points raised by the reviewers in the 'Response to Reviewers' box. Please attend to

all of the reviewers' comments. If you do not agree with any of their criticisms or suggestions please explain clearly why this is so.

Reviewer 1

Comments for the author

This appears to be a professionally conducted and well-written study revealing several clear correlations between body temperature and reproductive phases in female hamsters. Particularly this study appears to be the first searching for a relationship between litter size and body temperature. Indeed, a relation was found but it was too weak to predict litter size on body temperature variation only. The results should be useful for conservation-linked studies of hamsters and probably also other small mammals.

I only have a few minor comments:

Line 79. What does the ... in "litter size..." indicate?

Line 116 and legend to Fig. 1: When writing "preceding and following" please write that this refers to preceding and following parturition (if I got it right).

Comparing the temperatures recorded for reproducing females with non-reproducing females (Fig 3A vs 3B) reveals what to me looks like a considerably lower body temperature of the non-reproducing ones (e.g it never rising above 38 °C in 3B). I cannot find that the authors comment on this but I think it is well worth some consideration and may be of possible usefulness for field studies

Fig 5: Please indicate in the legend that this figure shows data from Gérard et al. (2025a) and not from the present study. Also explain what semi-captivity means, either in the legend or in the text (line 235-236).

Reviewer 2

Comments for the author

Major comments

This study investigated the use of body temperature (Tb) data from implanted temperature loggers to determine parturition events in the critically endangered common hamster (*Cricetus cricetus*). This technique could be applied in conservation practice to more accurately quantify reproductive output in small, cryptic mammals. Under controlled laboratory conditions, the authors found that mean daily Tb during parturition and the rate of recovery were linked to litter size. This information serves as a valuable validation step and provides a strong proof of concept for field application.

However, I expected this study to include field trials to validate its practical use in conservation settings. The laboratory experiments are a critical step in validating the technique, but field testing is necessary to determine whether it will work under real-world conditions. Many factors can compromise field deployment (e.g. equipment setup, weatherproofing, maintenance costs, vandalism, and overall practicality). If the authors are claiming that Tb monitoring is the most effective approach for assessing reproductive output for conservation purposes, then this needs to be demonstrated, at least in part, under field conditions.

Specific comments are provided below.

Abstract

An additional sentence explaining why body temperature is a reliable proxy for detecting parturition events helps readers understand the link. E.g. “elevated body temperature because...”

Introduction

The introduction is well written but would benefit from a tighter and more focused argument. In particular, the introduction should centre on endotherms, as there is no discussion of, or references to, ectotherms. Using broad terms such as “animals” or “species” therefore implies that the argument applies universally to animals, when in fact the examples and rationale are specific to mammals. For instance, the first half of the opening paragraph can remain broad, as the importance of wildlife monitoring and reproduction applies across taxa, but the second half should explicitly narrow the focus to mammals, where the subsequent monitoring examples and arguments are drawn.

I would also encourage the authors to emphasise the importance of accurate reproductive data for parameterising population dynamics and population viability models, which are commonly used to inform conservation and management decisions.

Methods

To ensure standardised practice and transparency in reporting methods and results from experimental biology research (Parker et al 2018), I've included a “Transparency, openness, and reproducibility” checklist at the end of the document from the Tools for Transparency in Ecology and Evolution (TTEE; <https://osf.io/g65cb/>) website. This is to warrant appropriate methodological details reported in the study. I do want to note that this checklist is generalised, and some of the items listed may not be applicable to this study.

Parker, T. H., Griffith, S. C., Bronstein, J. L., Fidler, F., Foster, S., Fraser, H., Forstmeier, W., Gurevitch, J., Koricheva, J. and Seppelt, R. (2018). Empowering peer reviewers with a checklist to improve transparency. *Nature Ecology & Evolution* 2, 929-936.

The description of the experiments is logical and well written, with most sections providing sufficient detail to address the study's aims. However, several areas require clarification:

1. **Validation:** What technique was used to validate that changes in Tb corresponded to reproductive events across all stages (pre-gestation, gestation, parturition, and post-parturition)? This information is not clearly stated in the manuscript. For example, were cameras used to monitor activity and behaviour, or was parturition checked daily by physically checking the enclosure at a standardised time? More generally, how can the authors be confident that the observed Tb trends are associated with parturition rather than other physiological or behavioural factors?
2. **Housing conditions:** What is meant by a “natural photoperiod” (L95)? How often were cages cleaned and maintained, and what bedding and substrate were provided?
3. **Animal impact:** If this technique is intended for field application in conservation contexts, an important consideration is the operational lifespan of the implanted loggers. Longer Tb time series would substantially increase the value of the data by allowing interpretation of daily, monthly, and seasonal variability, while also reducing the need for repeated recapture, which may raise ethical concerns. The expected logger lifespan should be stated explicitly and weighed against other existing approaches, such as changes in body mass or pup capture methods.

4. **Statistics:** Please provide more justification for the use of PCA with the collected data.

Depending on the objectives, a standard linear mixed-effects model may be sufficient. Clearly state the dependent and independent variables (e.g. female condition), specify which models were used for which analyses, and clarify that litter size was analysed using a GLM. Lastly, report the significance threshold used for all statistical analyses.

Results

Please also include effect estimates, test statistics, and degrees of freedom for all statistical results presented.

What is the likelihood that an increase in Tb is not associated with parturition? When considering the field application of this technique, it is important to assess the probability of false-positive detections of parturition based solely on Tb data. For example, in Figure 3B, some females that did not produce a litter still show increases in Tb. Visually, these trends appear similar (albeit weaker) than those observed in females that did undergo parturition (Figure 3A). This raises important questions about the reliability of using Tb alone to infer reproductive status. Is the PCA intended to distinguish true parturition events from other Tb increases?

For the paragraph in L149-156, the results are described in a way that largely reiterates the contents of the tables rather than highlighting the biological findings. A more informative approach would be to explicitly describe how each Tb metric differed among female reproductive statuses, including the direction and magnitude of the effects. For example, during parturition, females had, on average, an increase of X °C in Tb compared to females without parturition (with accompanying statistics).

I would also flesh out the PCA results more. Please explain what each principal component represents, the directionality of the loadings, and how females of different reproductive statuses are distributed within the multivariate space.

Finally, can the relationships between litter size and Tb plateau, mean Tb, and the steepness of Tb decline be visualised? Plotting these relationships would greatly aid interpretation and strengthen the presentation of the results.

Discussion

Some of the information presented in the first paragraph could be moved to the Results section. The opening of the Discussion should instead summarise the key findings without direct reference to tables or figures, focusing on the main take-home messages for the reader.

The Discussion would also benefit from further development of several arguments. First, a more in-depth interpretation of the relationship between Tb and parturition is needed. Referring broadly to energetics and metabolic rate provides only a high-level explanation, whereas multiple physiological processes could contribute to the observed increases in Tb during parturition. These could potentially include increased skeletal and smooth muscle activity, brown adipose tissue activation, and hormone-mediated changes in metabolism (e.g. thyroid hormones). There is an extensive experimental literature quantifying these mechanisms that are worth incorporating.

Second, the application of this technique for conservation monitoring requires more careful treatment. I had expected some form of field validation, and without such testing, conclusions about practical applicability remain limited. For this reason, I would be hesitant to state that *“The use of Tb variation is therefore a reliable and valuable method for*

identifying parturition dates and studying reproduction timing”. While the study provides a strong laboratory-based proof of concept, reliability cannot be claimed without field validation. I recommend rephrasing the Discussion to adopt a more cautious tone regarding the application of this method for conservation purposes.

Minor comments

L68: Explain what “*post-partum oestrus*” means.

L86: Provide some additional background on where the female common hamsters came from. Were they captive-bred or caught from the field? If from the field, where, when, and how were they captured? If captive-bred, how many generations (if known) in captivity, and who are the suppliers or breeders?

L89: How heavy were the loggers relative to the hamsters? <5% body weight? And how long were the data loggers kept inside?

L119: Why was 36 °C set as the threshold for defining torpor? L146: lower by how many degrees?

L207-209: Are there any studies linking parturition with increased energy expenditure? E.g. higher skeletal/smooth muscle activity, brown adipose tissue activity, hormone-induced changes in metabolism such as thyroid hormones, etc?

Figures

Figure 1: Include actual days in the x-axis.

Figure 3: How many females are represented in this figure?

Transparency openness, and reproducibility checklist

Category	Description	Checklist
Introduction		
Study purpose	State the original purpose for which the study was conducted and data were gathered	Yes
Methods		
Context	If paper is reporting results from a portion of a larger study, include a statement about the broader scope of the larger study and, if appropriate, indicate other publications from this study	N/A
Blinding	If possible, data recorders should be blind to the experimental treatment imposed on the subjects when gathering data. Also, report whether or not blinding was implemented.	No
Location	For field studies, include specific location(s) (e.g., latitude and longitude, elevation, water depth, habitat description)	N/A

Timing of study	Report study start date, end date, duration, and justification for duration and end date	No, only year.
Timing of sampling	Report timing (date, time of day if appropriate, etc.) and frequency of sampling, including storage duration for samples	Incomplete
Study conditions	Describe environmental or other conditions that may be relevant to the study question and taxa (e.g., temperature, light:dark cycle, etc.)	Incomplete
Subjects and treatments	Report methods used to choose subjects and to allocate subjects to treatments (e.g. randomized assignment), including organism taxon/taxa, source, and background (e.g., inbred lines, commercial seed, wild caught from X number of males and females and laboratory bred for Y generations, etc.) with institutional approvals as required and appropriate	Yes
Design	Describe design of experiment or study, including complete treatment factors and interactions, design structure (e.g., factorial, blocked, nested, hierarchical), nature of experimental units and replicates	Yes
Magnitude of treatment	Report both treatment and control values (with units and variation) for independent (explanatory/predictor) variables.	N/A
Sample size determination	Report how sample size was decided upon or determined. If sample size not set prior to initiation of study, explain stopping rule for sampling	No
Sample sizes	Report sample sizes for all data, including subsets of data (e.g., each treatment group, other subsets), and sample size used for all statistical analyses. Ideally also reported in results section.	Yes
Analysis methods	Provide the precise details of data analysis (including information on computer software programs and packages, and annotated full code or set of commands) as supplementary materials with submission and archived on a permanently supported platform upon publication	Yes
Data	Post data upon which analyses are based as supplementary materials with submission and archived in a permanently supported, publicly	No

		accessible database upon publication	
Materials		Stating appropriate materials throughout the methods, or provide comprehensive materials as supplementary documentation with submission and archived on a permanently supported platform upon publication . These are materials that are excluded from the methods section but which might be important for interpreting results or later attempts to replicate the study.	N/A
Voucher specimen		If relevant, possible and allowable, deposit voucher specimens of the studied taxon/taxa in an appropriate curated collection	N/A
Replication		If study is a replication, identify it as such and identify differences in methods between this study and original	N/A
Funding and conflicts of interest		Disclose all funding sources and potential conflicts of interest	Yes
Ethics and permit		Provide relevant details of ethical and other required permits if applicable (e.g., name of permit, permit number, etc.)	Yes
Results			
Complete reporting	statistical	List each statistical test and analysis conducted in sufficient detail such that they can be replicated and fully understood by those experienced in those methods. Fully report outcomes from each statistical analysis. For most analyses, this includes (but is not limited to) basic parameter estimates of central tendency (e.g., means) or other basic estimates (regression coefficients, correlation) and variability (e.g., standard deviation) or associated estimates of uncertainty (e.g., confidence/credible intervals) Thorough and transparent reporting will involve additional information that differs depending on the type of analyses conducted. For null hypothesis tests, this also should at minimum include test statistic, degrees of freedom, and p-value. For Bayesian analyses, this also should at a minimum include information on choice of priors and MCMC (Markov	Incomplete

	chain Monte Carlo) settings (e.g. burn-in, the number of iterations, and thinning intervals). For hierarchical and other more complex experimental designs, full information on the design and analysis, including identification of the appropriate level for tests (e.g. identifying the denominator used for split-plot experiments) and full reporting of outcomes (e.g. including blocking in the analysis if it was used in the design). Relevant information will differ among other types of analyses but in all cases should include enough information to fully evaluate the design and analysis	
post hoc acknowledgement	When hypotheses were formulated after data analysis, this should be acknowledged	N/A
References		
Citation of archived data, code, and materials	Properly cite any archived data, code, or materials made available by others and used in this manuscript	Yes

Reviewer's Responses to Questions

Experimental quality

Does each figure have the proper controls?

If 'No', please indicate reasons in Comments for Author box below.

Reviewer #1:

- Yes

Reviewer #2:

- Yes

Were the data analyzed using appropriate statistical tests?

If 'No', please indicate reasons in Comments for Author box below.

Reviewer #1:

- Yes

Reviewer #2:

- Yes

Reproducibility

Were experiments performed using adequate number of biological replicates?

If 'No', please indicate reasons in Comments for Author box below.

Reviewer #1:

- Yes

Reviewer #2:

- Yes

Does the methods section provide sufficient detail to permit reproducibility?

If 'No', please indicate reasons in Comments for Author box below.

Reviewer #1:

- Yes

Reviewer #2:

- No

Completeness

Are the manuscript's conclusions supported by the data?

If 'No', please indicate reasons in Comments for Author box below.

Reviewer #1:

- Yes

Reviewer #2:

- Yes

Scholarship

Do the authors cite and discuss the merits of data that would argue for and against their conclusion?

If 'No', please indicate reasons in Comments for Author box below.

Reviewer #1:

- Yes

Reviewer #2:

- Yes

Does the manuscript title & abstract accurately reflect the contents of the manuscript, without hyperbole?

If 'No', please indicate reasons in Comments for Author box below.

Reviewer #1:

- Yes

Reviewer #2:

- Yes

First revision

Author response to reviewers' comments

Detailed response to reviewers - Review round 1

REP: The authors would like to thank the two reviewers for their constructive comments.

Following the suggestion of reviewer 2, field data from a mesocosm monitoring were added to the article to support the claim that $T^{\circ}b$ signals can reliably indicate parturition in the wild. Because of this, two other collaborators are to be included as authors. Doing so, the usage of genetics, proposed in the discussion, was included in the article methods and results. This reshaped the article structure and figures numbering. We also addressed the various points of concerns raised by the reviewers in a point by point answers bellow. All changes are indicated in blue in the “Revised Manuscript - Highlighted Changes.docx” file (added text is underlined, removed text is crossed out). Point by point responses are detailed bellow in brown.

Reviewer 1

This appears to be a professionally conducted and well-written study revealing several clear correlations between body temperature and reproductive phases in female hamsters. Particularly this study appears to be the first searching for a relationship between litter size and body temperature. Indeed, a relation was found but it was too weak to predict litter size on body temperature variation only. The results should be useful for conservation-linked studies of hamsters and probably also other small mammals. I only have a few minor comments:

Line 79. What does the ... in "litter size..." indicate?

REP: We meant to say that the reproduction monitoring did not only focus on the timing of parturition, but also on other parameters, such as litter size. This detail was removed.

Line 116 and legend to Fig. 1: When writing "preceding and following" please write that this refers to preceding and following parturition (if I got it right).

REP: Actually, what we intended to convey here is that these parameters were calculated on a daily basis, independently of the female's physiological status at the time. Subsequently, they were evaluated against female status to determine their potential discriminatory power, as presented in Table 1 and the PCA (replaced by Figure 5).

This was made more explicit in Figure 1 legend as well as in line 121 and 126.

Comparing the temperatures recorded for reproducing females with non-reproducing females (Fig 3A vs 3B) reveals what to me looks like a considerably lower body temperature of the non-reproducing ones (e.g it never rising above $38^{\circ}C$ in 3B). I cannot find that the authors comment on this but I think it is well worth some consideration and may be of possible usefulness for field studies.

REP: (Figure 3 was renumbered into Figure 4)

Though it may seem that way in Figure 4, non-reproducing females did not have a lower $T^{\circ}b$ than the ones that reproduced when they were outside of the reproductive period (GLM, $p = 0.097$). The differences observed on Figure 4 is to be attributed to reproduction, with increased body T° already in a presence of a male, then during gestation, and finally during pups rearing. On graph 4.A, considering a baseline of ($-37.6^{\circ}C$) you can see $T^{\circ}b$ increasing from day -22 (corresponding to male's arrival) and returning to baseline at -25 days (after 3.5 weeks of pups rearing). On graph B, non-reproducing females' $T^{\circ}b$ stayed at an average baseline from mid-May to mid-July. The lower mean temperature before and after this range is hard to interpret, as it is mainly due to 2 individuals showing lower $T^{\circ}b$, which could be considered as noise.

The idea of $T^{\circ}b$ being more elevated throughout reproduction has been explicitly added in the Discussion Line 258-260

Fig 5: Please indicate in the legend that this figure shows data from Gérard et al. (2025a) and not from the present study. Also explain what semi-captivity means, either in the legend or in the text (line 235-236).

REP: (Figure 5 was renumbered into Figure 3)

Because Mesocosm monitoring has been added as a whole part in this second version of the article, the data for Figure 3 is now included in the text.

Reviewer 2

This study investigated the use of body temperature (T_b) data from implanted temperature loggers to determine parturition events in the critically endangered common hamster (*Cricetus cricetus*). This technique could be applied in conservation practice to more accurately quantify reproductive output in small, cryptic mammals. Under controlled laboratory conditions, the authors found that mean daily T_b during parturition and the rate of recovery were linked to litter size. This information serves as a valuable validation step and provides a strong proof of concept for field application.

However, I expected this study to include field trials to validate its practical use in conservation settings. The laboratory experiments are a critical step in validating the technique, but field testing is necessary to determine whether it will work under real-world conditions. Many factors can compromise field deployment (e.g. equipment setup, weatherproofing, maintenance costs, vandalism, and overall practicality). If the authors are claiming that T_b monitoring is the most effective approach for assessing reproductive output for conservation purposes, then this needs to be demonstrated, at least in part, under field conditions.

REP: Thank you for this very helpful suggestion. Our initial decision was to present only the laboratory validation of the method, as we felt that it most clearly highlighted the central concept that T_b can be used to detect parturition. However, we agree that extrapolation to free-ranging conditions requires empirical support under field settings. As part of a separate research program conducted over 3 years, we monitored T_b in reproducing females maintained under semi-natural (enclosure) conditions. Although these data were originally collected for different research objectives, they provide valuable support for the applicability of the method beyond the laboratory context. Following your recommendation, we have now included these additional data into the manuscript and expanded the discussion accordingly. We hope that this addition adequately addresses your concern.

Specific comments are provided below.

Abstract

An additional sentence explaining why body temperature is a reliable proxy for detecting parturition events helps readers understand the link. E.g. “elevated body temperature because...”

REP: Statement added line 17-18.

Introduction

The introduction is well written but would benefit from a tighter and more focused argument. In particular, the introduction should centre on endotherms, as there is no discussion of, or references to, ectotherms. Using broad terms such as “animals” or “species” therefore implies that the argument applies universally to animals, when in fact the examples and rationale are

specific to mammals. For instance, the first half of the opening paragraph can remain broad, as the importance of wildlife monitoring and reproduction applies across taxa, but the second half should explicitly narrow the focus to mammals, where the subsequent monitoring examples and arguments are drawn.

REP: Focus was narrowed with changes in Lines 45, 47, 48, 50 and 52.

I would also encourage the authors to emphasise the importance of accurate reproductive data for parameterising population dynamics and population viability models, which are commonly used to inform conservation and management decisions.

REP: This was added Line 42-43

Methods

To ensure standardised practice and transparency in reporting methods and results from experimental biology research (Parker et al 2018), I've included a "Transparency, openness, and reproducibility" checklist at the end of the document from the Tools for Transparency in Ecology and Evolution (TTEE; <https://osf.io/g65cb/>) website. This is to warrant appropriate methodological details reported in the study. I do want to note that this checklist is generalised, and some of the items listed may not be applicable to this study.

Parker, T. H., Griffith, S. C., Bronstein, J. L., Fidler, F., Foster, S., Fraser, H., Forstmeier, W., Gurevitch, J., Koricheva, J. and Seppelt, R. (2018). Empowering peer reviewers with a checklist to improve transparency. *Nature Ecology & Evolution* 2, 929-936.

The description of the experiments is logical and well written, with most sections providing sufficient detail to address the study's aims. However, several areas require clarification:

1. Validation: What technique was used to validate that changes in Tb corresponded to reproductive events across all stages (pre-gestation, gestation, parturition, and post-parturition)? This information is not clearly stated in the manuscript. For example, were cameras used to monitor activity and behaviour, or was parturition checked daily by physically checking the enclosure at a standardised time? More generally, how can the authors be confident that the observed Tb trends are associated with parturition rather than other physiological or behavioural factors?

REP: Parturition was recorded through direct visual inspection of the cages. Cage checks were conducted daily (rephrased, Line 116). We did not collect behavioural data. Our objective was to identify parturition using a monitoring approach that could be realistically implemented under field conditions, with visual confirmation serving as validation of the method.

2. Housing conditions: What is meant by a "natural photoperiod" (L95)? How often were cages cleaned and maintained, and what bedding and substrate were provided?

REP: The term "natural photoperiod" was intended to indicate that artificial lighting mimicked the natural daylight cycle, with daily variation in day length throughout the year. This has been clarified (Line 106). Details regarding substrate, cage maintenance and enrichment have also been added Line 106 to 109.

3. Animal impact: If this technique is intended for field application in conservation contexts, an important consideration is the operational lifespan of the implanted loggers. Longer Tb time series would substantially increase the value of the data by allowing interpretation of daily, monthly, and seasonal variability, while also reducing the need for repeated recapture, which

may raise ethical concerns. The expected logger lifespan should be stated explicitly and weighed against other existing approaches, such as changes in body mass or pup capture methods.

REP: Logger limitations and T°b monitoring advantage compared to other methods is now further discussed line 285-293, and 302 - 307.

4. Statistics: Please provide more justification for the use of PCA with the collected data. Depending on the objectives, a standard linear mixed-effects model may be sufficient. Clearly state the dependent and independent variables (e.g. female condition), specify which models were used for which analyses, and clarify that litter size was analysed using a GLM. Lastly, report the significance threshold used for all statistical analyses.

REP: The statistical analysis section has been thoroughly revised (Line 174 to 189) and the Supplementary Tables have been reorganized for clarity.

The original purpose of the PCA was to jointly represent daily T°b metric variability, T°b metric correlation and their relationship with female physiological status. However, to improve clarity, we ultimately chose to focus on $\mu T^{\circ}b$ and $\Delta T^{\circ}b$, as these variables provided the strongest discrimination of parturition. Accordingly, the PCA has been replaced with a scatterplot including confidence intervals.

Regarding the Transparency openness, and reproducibility checklist where missing information where highlighted :

Blinding: If possible, data recorders should be blind to the experimental treatment imposed on the subjects when gathering data. Also, report whether or not blinding was implemented.

REP: Blinding was not possible, as no treatments were imposed to the subjects.

Timing of study: Report study start date, end date, duration, and justification for duration and end date

REP: Starting date was added line 115. End of monitoring depended of female's treatment, as explained Lines 128-129.

Timing of sampling: Report timing (date, time of day if appropriate, etc.) and frequency of sampling, including storage duration for samples.

REP: Litter observation timing was added line 124.

Study conditions: Describe environmental or other conditions that may be relevant to the study question and taxa (e.g., temperature, light:dark cycle, etc.)

REP: Provided Line 96 to 112.

Data: Post data upon which analyses are based as supplementary materials with submission and archived in a permanently supported, publicly accessible database upon publication

REP: Stated Line 191-192 and 340 - 344.

Results

Please also include effect estimates, test statistics, and degrees of freedom for all statistical results presented.

REP: The relevant information has been added into the main text, except for the models investigating the relationship between $T^{\circ}b$ characteristics and female status, which would have resulted in excessive detail within the main article body. Instead, the full model outputs and pairwise comparisons were provided in the Supplementary Material, while only the key statistics are reported in the main text. For the remaining models, because details are now provided in the main body of the manuscript, the previously redundant supplementary tables have been removed.

What is the likelihood that an increase in Tb is not associated with parturition? When considering the field application of this technique, it is important to assess the probability of false-positive detections of parturition based solely on Tb data. For example, in Figure 3B, some females that did not produce a litter still show increases in Tb. Visually, these trends appear similar (albeit weaker) than those observed in females that did undergo parturition (Figure 3A). This raises important questions about the reliability of using Tb alone to infer reproductive status. Is the PCA intended to distinguish true parturition events from other Tb increases?

REP: It is true that non-reproducing females could also show punctual increase in $T^{\circ}b$, but these are weaker ($<0.5^{\circ}C$ in amplitude) and do not lead to $T^{\circ}b$ exceeding $38.5^{\circ}C$. Thus, they can be discriminated from parturition signals, as added Line 216-217. The reliability of this signal has also been further discussed Line 294-295.

For the paragraph in L149-156, the results are described in a way that largely reiterates the contents of the tables rather than highlighting the biological findings. A more informative approach would be to explicitly describe how each Tb metric differed among female reproductive statuses, including the direction and magnitude of the effects. For example, during parturition, females had, on average, an increase of $X^{\circ}C$ in Tb compared to females without parturition (with accompanying statistics).

REP: The result section was rewritten accordingly.

I would also flesh out the PCA results more. Please explain what each principal component represents, the directionality of the loadings, and how females of different reproductive statuses are distributed within the multivariate space.

REP: For more clarity, the PCA was replaced by a Graph focusing on the two parameters that were the most specific to parturition days: $\mu T^{\circ}b$ and $\Delta T^{\circ}b$ (Figure 5).

Finally, can the relationships between litter size and Tb plateau, mean Tb, and the steepness of Tb decline be visualised? Plotting these relationships would greatly aid interpretation and strengthen the presentation of the results.

REP: Added for both laboratory and mesocosm monitorings as Figure 6.

Discussion

Some of the information presented in the first paragraph could be moved to the Results section. The opening of the Discussion should instead summarise the key findings without direct reference to tables or figures, focusing on the main take-home messages for the reader.

REP: The discussion was rewritten, and the message of the top part was narrowed.

The Discussion would also benefit from further development of several arguments. First, a mo

re in- depth interpretation of the relationship between Tb and parturition is needed. Referring broadly to energetics and metabolic rate provides only a high-level explanation, whereas multiple physiological processes could contribute to the observed increases in Tb during parturition. These could potentially include increased skeletal and smooth muscle activity, brown adipose tissue activation, and hormone - mediated changes in metabolism (e.g. thyroid hormones). There is an extensive experimental literature quantifying these mechanisms that are worth incorporating.

REP: It is correct that we did not cite literature addressing the physiological mechanisms underlying the increase in $T^{\circ}b$. Instead, we intentionally focused on the potential adaptive benefits of this elevation. Our rationale is that, if increased $T^{\circ}b$ confers reproductive advantages, similar benefits may be expected across small mammal species, suggesting that the pattern observed in the common hamster could extend to other taxa. Accordingly, we considered that an in-depth discussion of the underlying mechanisms might divert attention from the central message of the manuscript rather than reinforce it.

Second, the application of this technique for conservation monitoring requires more careful treatment. I had expected some form of field validation, and without such testing, conclusions about practical applicability remain limited. For this reason, I would be hesitant to state that “The use of Tb variation is therefore a reliable and valuable method for identifying parturition dates and studying reproduction timing”. While the study provides a strong laboratory-based proof of concept, reliability cannot be claimed without field validation. I recommend rephrasing the Discussion to adopt a more cautious tone regarding the application of this method for conservation purposes.

REP: Mesocosm monitoring was added to provide field validation throughout the article, as discussed previously.

Minor comments

L68: Explain what “post-partum oestrus” means.

REP: Done line 78

L86: Provide some additional background on where the female common hamsters came from. Were they captive-bred or caught from the field? If from the field, where, when, and how were they captured? If captive-bred, how many generations (if known) in captivity, and who are the suppliers or breeders?

REP: Added line 97-98

L89: How heavy were the loggers relative to the hamsters? <5% body weight?

REP: Added Line 100 (~1.5% of adults body mass)

And how long were the data loggers kept inside?

REP: Added Line 102 103

L119: Why was 36 °C set as the threshold for defining torpor?

REP: This is because torpors (i.e. rapid and pronounced declines in $T^{\circ}b$) strongly biased the different $T^{\circ}b$ metrics. Consequently, they had to be excluded from the analyses, even when they

were relatively shallow. For this reason, we applied a high threshold for exclusion, as now specified in lines 137-138.

L146: lower by how many degrees?

REP: We were unable to find what this question refers to in the original text line 146.

L207-209: Are there any studies linking parturition with increased energy expenditure? E.g. higher skeletal/smooth muscle activity, brown adipose tissue activity, hormone-induced changes in metabolism such as thyroid hormones, etc?

REP: See response to the main comment about this above.

Figures

Figure 1: Include actual days in the x-axis.

REP: Done

Figure 3: How many females are represented in this figure?

REP: (Now figure 4) Figure 4.A: n = 15. Figure 4.B: n = 4. Stated in the legend.

Second decision letter

MS ID#: bio.062459R1

MS Title: Monitoring reproduction in cryptic small mammals; Using body temperature to identify parturition in an endangered rodent

Authors: Timothée Gérard, Hugo Chignec, Chantal Poteau, Emilie Long and Caroline Habold

I have read through your revised manuscript, am happy to tell you that your manuscript has now been accepted for publication in Biology Open, pending our standard publication integrity checks. It was accepted on 19th March 2026. In particular I appreciate the additional experiments that you have added to the manuscript.